# Utilizing Entropy-Based Method for Rainfall Network Design in Huaihe River Basin, China

**Jian Liu** [1,2,*], **Yanyan Li** [1,2], **Yuankun Wang** [3] and **Pengcheng Xu** [4]

[1] Water Resources Research Institute of Shandong Province, Jinan 250014, China; liyanyan026@outlook.com
[2] Shandong Province Key Laboratory of Water Resources and Environment, Jinan 250014, China
[3] School of Water Resources and Hydropower Engineering, North China Electric Power University, Beijing 102206, China; yuankunw@ncepu.edu.cn
[4] College of Hydraulic Science and Engineering, Yangzhou University, Yangzhou 225008, China; m18994113495@163.com
* Correspondence: water_liujian@163.com

**Abstract:** The nonstationary characteristics caused by significant variation in hydrometeorological series in the context of climate change inevitably have a certain impact on the selection of an optimal gauging network. This study proposes an entropy-based, multi-objective, rain gauge network optimization method to facilitate the design of a 43 stations-based network in Huaihe River Basin (HRB), China. The first goal of this study is to improve the accuracy of gauge-related information estimation through the selection and comparison of discretization methods. The second goal of this study is to quantify the impact of trend-caused nonstationarity on optimal network design using the sliding window method. This study compares the divergence of three kinds of discretization methods, including the floor function-based approach, Scott's equal bin width histogram (EWH-Sc) approach, and Sturges's equal bin width histogram (EWH-St) approach. The matching degree of the variance and marginal entropy of the observed series is computed to select the most suitable of the above three discretization methods. The trend-caused nonstationarity in 75% of all stations in the HRB could definitely influence the final results of the optimal rain-gauge network design using the sliding window method. Therefore, in future studies of rain-gauge network optimization, it is necessary to carry out uncertainty research according to local conditions in view of climate change and human activities.

**Keywords:** rain gauge network; nonstationarity; multi-objective problems; climate change

## 1. Introduction

The key elements, including rainfall, runoff, sediment, water quality, and the water environment, of the hydrological cycle and water resource management activities require the support of hydrological observation activities. The smooth and efficient operation of water resource planning and management requires the scientific and reasonable spatial distribution of hydrometeorological gauge networks [1,2]. The intensification of human activities in the context of climate change has altered the current situation of hydrological cycles in watersheds. Such alterations affect key hydrological and meteorological factors and subtly change the evolution process of hydrometeorological cycles within the watershed system [3]. The systematic planning and deployment of hydrometeorological gauge networks is clearly large-scale work that involves long-term interests. Although their short-term benefits are insignificant, they are significantly beneficial in solving water supply conflicts, optimizing reservoir scheduling, optimizing water resource allocation, and smoothly implementing agricultural irrigation mechanisms [4].

Optimization of a gauge-based network often starts with statistical regression analysis to carry out real-time positioning, gauge-based research. The generalized least squares (GLS) approach is a typical regression analysis method that maximizes regional information

within a limited budget and timeframe. In terms of spatial differences in station distribution, geostatistical methods also have significant effects in optimizing hydrological gauge networks [5]. The universality of the kriging-based geostatistical method is mainly reflected in the following two levels [6,7]: (1) the first level is based on the variation function for spatial interpolation analysis, and (2) the second level lies in the reasonable allocation of weight coefficients. Shafiei et al. [8] also proposed that efficient station network design depends on accurate rainfall prediction rates at the minimum interpolation level. Due to its built-in, good uncertainty matrix, the copula-based spatial interpolation method not only contains model attributes and positioning information, but it also comprehensively considers information related to observation samples [9].

Information entropy theory has been widely applied in the field of hydrometeorology since the 1970s [10]. The core idea of rain gauge network optimization using information entropy is to reduce the amount of information transfer between stations as much as possible. Multiple entropy-based indicators, including trans-information, joint entropy, and total correlation, have been utilized in the development and design of hydrometeorological gauge networks. The directional information transfer (DIT) index has been proposed for the design of rainfall and runoff networks [11]. The total correlation index introduced by Alfonso et al. [12] extends the measurement of trans-information to multivariate situations and promotes multi-objective optimization of rain gauge network design. The joint entropy and total correlation indexes quantify the information content and redundancy, respectively. The weight factors of the above two objective functions can be obtained from the Pareto solution through a greedy algorithm. Considering the sensitivity of entropy-based information metrics to discretization methods, the selection of discretization methods is always the key to restricting the optimal station network scheme in the process of station network optimization. Based on the information entropy-utility standard, Fuentes et al. [13] constructed a gauge network spatial evaluation model under the non-stationary framework and evaluated pertinent environmental and air pollution samples. Li et al. [14] combined entropy- and copula-based methods to derive the optimal rain gauge network in Taihu Lake Basin. In order to augment ungauged areas with new stations, they developed a high-value-of-monitoring index to systematically design the gauge network.

In the design and management of hydrometric gauge networks, climate change and human activities are potential factors causing nonstationarity in hydrological processes. The trend-caused nonstationarity cannot be ignored, especially for rainfall events that are highly heterogeneous, localized, and influenced by geographical, topographical, and climatic factors [15–20]. In the context of the scarcity of observed data, this study intends to discuss differences in station network optimization results under different time scales and periods, so as to further explore the impact of temporal variability on the optimization of hydrometric networks.

The goals of this study are stated from two angles: (1) comprehensively compare the impacts of three information entropy discretization estimation methods on gauge optimization results, and select the optimal discretization method; (2) apply the information entropy-based, multi-objective optimization algorithm and sliding window method to analyze the optimization of a hydrometric network in the Huaihe River Basin, China. Furthermore, the impacts of temporal variability in rainfall series on the optimal results are discussed in this study.

## 2. Study Area and Dataset

### 2.1. Study Area

The Huaihe River Basin (HRB) is located in the eastern part of China (Figure 1). It starts from the Tongbai and Funiu Mountains to the west, faces the Yellow Sea to the east, and is bordered by the Dabie Mountains, Jianghuai Hills, Tongyang Canal, and Rutai Canal to the south and the Yangtze River Basin to the north. The HRB is located in the climate transition zone between the north and south of China. The north of the HRB belongs to the warm, temperate, semi-humid monsoon climate zone, while the south of the HRB belongs

to the subtropical, humid monsoon climate zone. The basin has a transitional climate type from warm temperate to subtropical from north to south, with frequent cold and warm air masses and significant changes in precipitation. The distribution of the annual precipitation is uneven, with the rainy season concentrated from May to September in the upper reaches of the HRB and the Huainan mountainous areas, while it is concentrated from June to September in other regions.

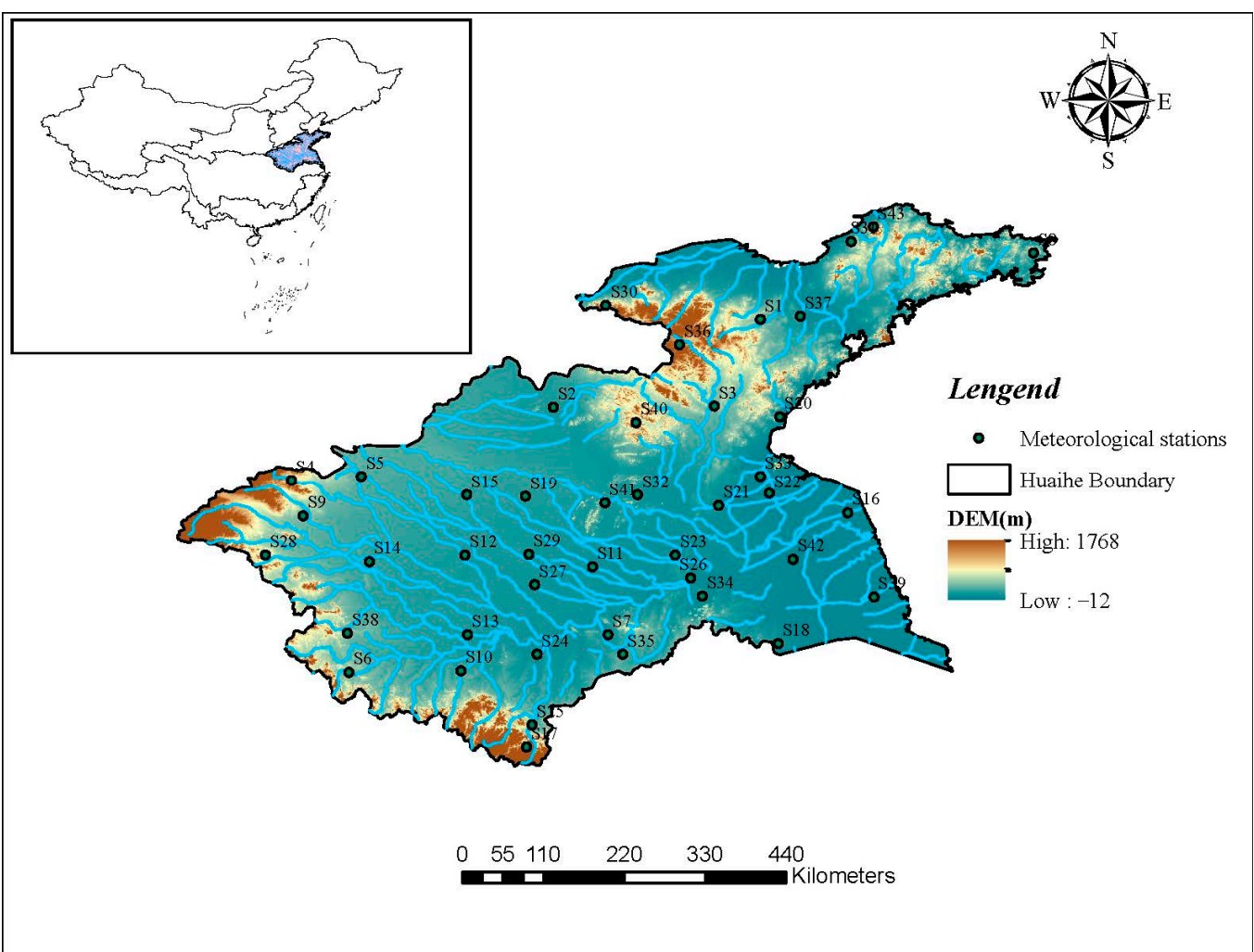

**Figure 1.** Study area: Huaihe River Basin with 43 meteorological stations. DEM, digital elevation model.

With the acceleration of urbanization in the HRB and dual impact of climate change, the frequent activities of the rainstorm weather system cause floods and waterlogging disasters, which are becoming the main bottleneck of national economic development. The present hydrometeorological gauge network composed of 43 stations needs to be analyzed and updated to help provide real-time forecasting and data support for extreme rainstorm and flood risk management.

*2.2. Data Processing*

In this study, daily precipitation data during 1968–2018 in the HRB was used for rain gauge network design. Considering the impacts of data structure autocorrelation on the assumption of independent and identically distributed, we proposed a data fusion strategy to alleviate sequence autocorrelation as follows:

(i)　Firstly, regional rainy days were selected as at least one station having precipitation records ($> 0$ mm). In contrast, regional non-rainy days were defined as all stations in the basin having no precipitation records.

(ii)　Secondly, adjacent regional rainy days would be separated by non-rainy days. Thus, effective interval days ($n_{EI}$) were defined, which was used to accumulate precipitation records from adjacent rainy days.

(iii)　Thirdly, autocorrelation tests were implemented to ensure the processed series obeyed the "independent and identically distributed" assumption.

The $n_{EI}$ value should be appropriate to help eliminate the autocorrelation and make sure entropy is calculated. An example with $n_{EI} = 3$ is shown in Figure 2. After processing, the series length was shortened from 18,628 to 605.

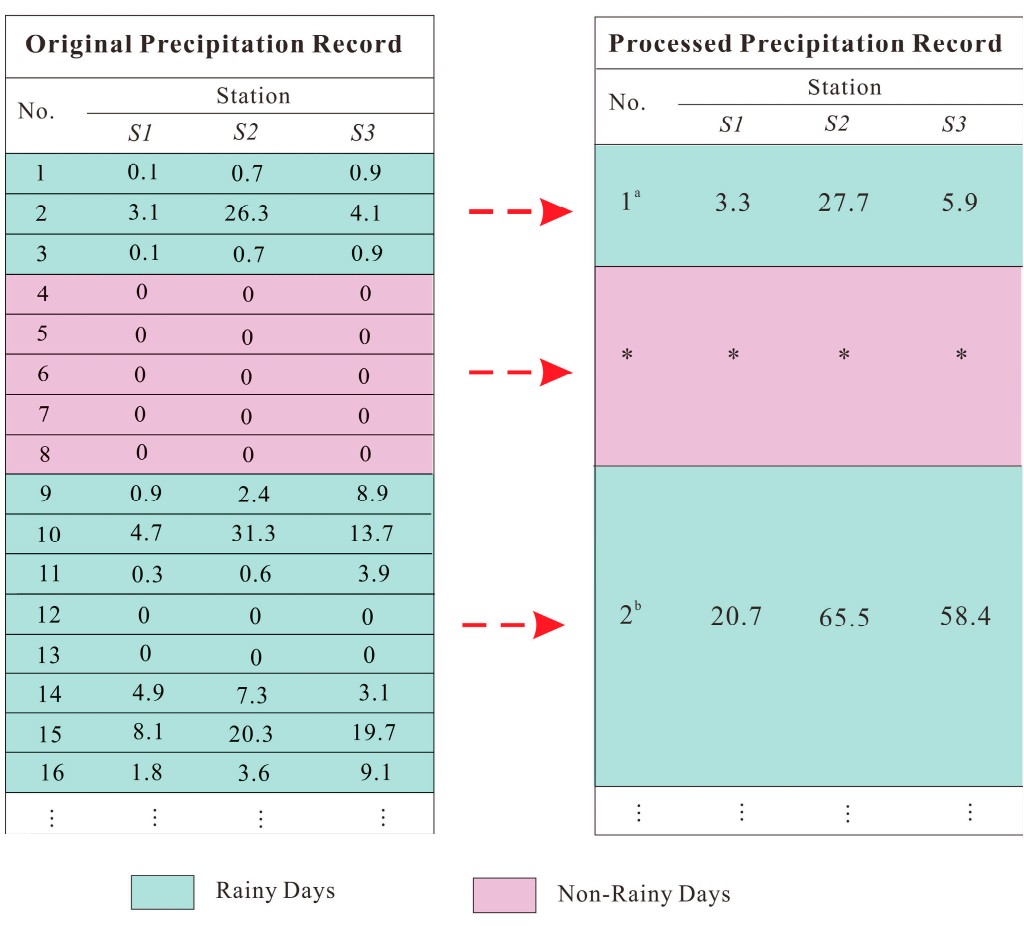

**Figure 2.** Explanation of data fusion process with effective interval days ($n_{EI}$) being 3.

## 3. Methods

### 3.1. Entropy-Based Indexes for Station Network Optimization

Entropy can be a remarkable metric to quantify the potential uncertainty of hydrometeorological variables, such as precipitation or streamflow, which can represent the information transferred by a gauging network [21]. In other words, if we assume that rainfall events are a random process, information entropy can be applied to quantify the uncertainty of rainfall information based on the rainfall time series.

Let $[X_1, X_2, \ldots, X_d]$ represent a set of $d$-dimensional random variable samples (a gauging network consisting of $d$ corresponding stations in this study). Here, $X_1, X_2, \ldots, X_d$ represent the processed precipitation series extracted from these $d$ stations. Its joint probability density function is $p(x_1, x_2, \ldots, x_d)$. $p_{X_1}(x_1), p_{X_2}(x_2), \ldots,$ and $p_{X_d}(x_d)$ denote the corresponding probability density function (PDF) of each variable, respectively. The three kinds of commonly used entropy-based indexes are marginal entropy, joint entropy, and total correlation. Marginal entropy ($H(X_i)$) represents the total information contained in a single variable, while joint entropy ($H(X_1, X_2, \ldots, X_d)$) describes the total information contained in multiple variables. The total correlation ($TC(X_1, X_2, \ldots, X_d)$) represents the amount of redundant information between multiple variables. They can be specifically defined as:

(1)　Marginal entropy (*ME*):

$$H(X_i) = -\sum\nolimits_{x_i} p(x_i) log_2 p(x_i) \tag{1}$$

(2)　Joint entropy (*JE*):

$$H(X_1, X_2, \ldots, X_d) = -\sum_{x_1} \cdots \sum_{x_d} p(x_1, x_2, \ldots, x_d) log_2(p(x_1, x_2, \ldots, x_d)) \tag{2}$$

(3)　Total correlation (*TC*):

$$TC(X_1, X_2, \ldots, X_d) = \sum_{i=1}^{d} H(X_i) - H(X_1, X_2, \ldots, X_d) \tag{3}$$

As shown in the above equations, the *JE* value is adopted in network optimization as a measure of total network information while the *TC* value is adopted as a metric of information redundancy.

Inspired by the multi-objective optimization approach proposed by Alfonso et al. [12], this study adopted the criterion of maximizing the joint entropy and minimizing the total correlation (max*JE*-min*TC*), which can be exhibited as follows:

$$\begin{cases} maxF_1 = \max\{H(X_{S_1}, X_{S_2}, \ldots, X_{S_m})\} \\ minF_2 = \min\{TC(X_{S_1}, X_{S_2}, \ldots, X_{S_m})\} \end{cases} \tag{4}$$

Here, $X_{S_1}, X_{S_2}, \ldots,$ and $X_{S_m}$ represent the precipitation series from the selected $m$ stations out of the original $d$ stations ($m \leq d$). Based on these two objective functions, the best gauge combination for the target network can be determined by maximizing the total network information and minimizing the network redundancy. These two objective functions are commonly used in optimizing hydrological gauging networks, and the criterion used in this study followed this approach.

Here, we utilized the example of the 43 stations in this study (Figure 3). Initially, the original network comprised 43 stations, which were assigned numbers S1 to S43. We obtained the best gauge combination from the following steps:

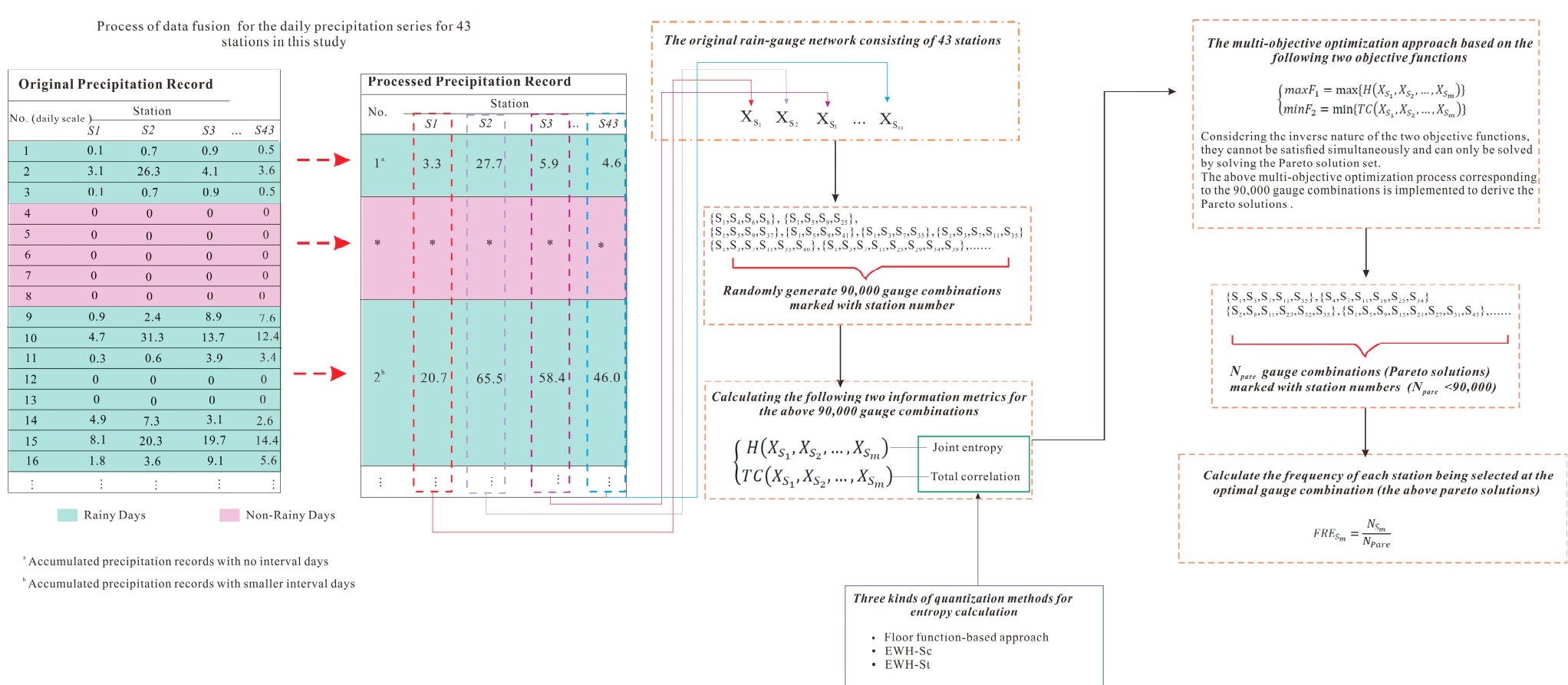

**Figure 3.** The flowchart of this study (* Deleted records with bigger interval days).

(1)　For example, choosing 2 sites from 43 sites would result in 903 ($C_{43}^2 = \frac{43!}{41! \times 2!} = 903$) possible combinations. In the same way, choosing 6 sites from 43 sites would result in 6,096,454 ($C_{43}^6 = \frac{43!}{37! \times 6!} = 6{,}096{,}454$) possible combinations. Therefore, it was not necessary to exhaustively search all possible combinations of the given number of gauges. Instead, we commenced by generating a multitude of potential gauge combinations, specifically 90,000 in this study.

(2)　The joint entropy and total correlation for each combination of gauges can be calculated using the processed precipitation data. However, it is important to note that these two information indexes, namely joint entropy and total correlation, need to be computed using an appropriate discretization method, which is investigated in Section 3.2.

(3)　Based on these two objective functions, ($\max\{H(X_{S_1}, X_{S_2}, \ldots, X_{S_m})\}$ and $\min\{TC(X_{S_1}, X_{S_2}, \ldots, X_{S_m})\}$), a certain number of Pareto solutions ($N_{Pare}$) can be derived as approximations to the optimal function. This is because it is impossible to satisfy both objective functions simultaneously.

(4)　The frequency of station selection is calculated by examining the occurrence of the label (in this case, $S_m$ represents the label for a specific station) in the Pareto solutions. Since different stations have different frequencies of label occurrence, the selected frequency can be determined using the following calculation:

$$FRE_{S_m} = \frac{N_{S_m}}{N_{Pare}} \tag{5}$$

where $N_{S_m}$ denotes the number of occurrences of each corresponding station in the Pareto solutions. $FRE_{S_m}$ is the selection frequency of a station.

### 3.2. Three Kinds of Quantization Methods for Entropy Calculation

The above entropy-based metrics are calculated in the discrete form rather than the continuous form. In this study, three quantization approaches with fixed or dynamic bin widths were proposed to calculate the discrete entropy: the floor function-based approach, Scott's equal bin width histogram (EWH-Sc) approach, and Sturges's equal bin width histogram (EWH-St) approach. Data discretization refers to dividing continuous data into discrete intervals. In this study, the daily processed precipitation series of each station were continuous data and needed to be divided into several discrete intervals using the aforementioned approaches. The widths of the intervals were determined based on the discretization approach. The joint entropy and total correlation in Equation (4), which are two objective functions, must be calculated based on discrete distribution instead of continuous distribution. The daily precipitation series should be discretized accordingly.

The floor function-based approach can be regarded as a rounding method to transform the original data through a constant, $a$, which can be defined as:

$$x_q = a \left\lfloor \frac{2x + a}{2a} \right\rfloor \tag{6}$$

where $x$ is the observed sample series; $x_q$ is discretized data; $\lfloor \cdot \rfloor$ is the rounding down function; and $a$ is the key preset parameter for the floor function-based approach. In this study, we set $a$ as 50, 100, 150, and 1000 for further analysis. The floor function-based approach can convert the precipitation variable to the nearest lowest integer multiples of constant a through a mathematical floor function.

EWH-Sc was proposed by Scott [22] to optimize bin width by minimizing the difference between true and histogram-estimated density values. The optimal bin width ($a_{\text{EWH-Sc}}$) can be computed as follows:

$$a_{\text{EWH-Sc}} = 3.49 \sigma_X N^{-\frac{1}{3}} \tag{7}$$

where $\sigma_X$ is the sample standard deviation. $N$ is the sample length.

Sturges's equal bin width histogram (EWH-St) approach [23] can be implemented as follows:

$$a_{\text{EWH}-\text{St}} = \frac{R_x}{1 + \log_2 N} \tag{8}$$

where $R_x$ denotes the difference between the maximum and minimum values of sample $X$.

## 4. Results

### 4.1. Analysis of Processed Data in the HRB

Due to the periodic nature of hydrometeorological series, they unavoidably have an impact on the autocorrelation of hydrometeorological series, thereby violating the assumption of independent and identically distributed (*i.i.d*) hydrometeorological sequences. This assumption is crucial as it provides the basis for the randomness and independence of hydrometeorological variables. In order to address this issue, this study employed a data fusion method that involved time interval partitioning (please refer to Section 2.2 for a detailed explanation of the implementation process) in order to ensure that the processed data's autocorrelation met the assumption of being independent and identically distributed. We utilized autocorrelation function-based tests (ACF) to assess the extent to which the processed data satisfied the *i.i.d* assumption. Due to space limitations, we only used data from 16 stations, labeled S1–S16, to illustrate the contrast between the ACF values of the original data and the processed precipitation series. As shown in Figure S1a, almost all of the ACF values, represented by a black vertical line for processed data extracted from all selected 16 stations, were between the two blue dashed lines, suggesting that the processed data with $n_{EI}$ being 3 could adhere to the *i.i.d* assumption. In contrast, the original daily precipitation failed the ACF test, as the ACF values exceeded the blue dashed lines (Figure S1b). It can be observed that the accumulation processing of the series not only weakened the autocorrelation of the sequence itself, but it also increased the correlation between station pairs (the correlation coefficient was close to 1.0 in Figure S2a, while the correlation coefficient in Figure S2b was significantly smaller than that in Figure S2a). Furthermore, the statistical characteristics (Table 1), including the maximum and standard deviation of the processed data, were expected to be larger than those of the original data. This was because the processed data were generated by accumulating precipitation records from adjacent rainy days in Section 2.2. Specifically, the maximum value obtained from the processed data increased by almost 8–9 times compared to the original data, while the standard deviation value derived from the processed data increased by approximately 18–19 times compared to the original data.

**Table 1.** The detailed characteristics of the grouped precipitation data derived from the 16 selected stations.

| Station No. | Original Data | | Processed Data | |
|:---:|:---:|:---:|:---:|:---:|
| | Maximum (mm) | SD [a] (mm) | Maximum | SD |
| S1 | 188.8 | 6.9 | 945.7 | 132.9 |
| S2 | 191.3 | 8.4 | 868.6 | 151.2 |
| S3 | 288.6 | 8.9 | 1183.7 | 173.3 |
| S4 | 189.4 | 7.4 | 818.4 | 131.1 |
| S5 | 217.8 | 7.7 | 913.7 | 134.4 |
| S6 | 276.2 | 10.9 | 1279.3 | 210.3 |
| S7 | 216.7 | 9.7 | 1086.3 | 182.8 |
| S8 | 257.7 | 8.6 | 1073.9 | 144.5 |
| S9 | 177.2 | 7.9 | 896.9 | 145.4 |
| S10 | 206.9 | 10.5 | 1313.7 | 104.4 |
| S11 | 232.6 | 9.3 | 1193.9 | 172.4 |

**Table 1.** *Cont.*

| Station No. | Original Data | | Processed Data | |
|---|---|---|---|---|
| | Maximum (mm) | SD [a] (mm) | Maximum | SD |
| S12 | 285.3 | 9.1 | 1013.3 | 161.3 |
| S13 | 226.1 | 9.8 | 1186.8 | 181.1 |
| S14 | 225.4 | 8.8 | 1109.7 | 156.2 |
| S15 | 363.6 | 8.6 | 1062.9 | 149.6 |
| S16 | 263.2 | 9.9 | 1456.0 | 198.3 |

Note: [a] SD represented standard deviation.

*4.2. Selection of Data Discretization Method*

Since data discretization methods have a significant impact on the results of station network optimization, it was necessary to analyze the differences between three discretization methods when calculating information quantitative indicators. To account for the large number of gauge combinations, we calculated the joint information entropy (*JE*) and total correlation (*TC*) for 30 potential gauge combinations. The calculation results are presented in Figure 4. It is evident from Figure 4 that the estimated joint information entropy and total correlation obtained using the Scott discretization method were greater than those calculated using the other two discretization methods. Furthermore, upon examining Figure 4, it was observed that when applying the floor function-based approach to compare the information entropy calculations for different values of parameter *a*, the trends of *JE* and *TC* were approximately similar. Specifically, as the value of *a* increased, the values of *JE* and *TC* decreased. Therefore, when the value of *a* was larger, both the *JE* and *TC* values were smaller. This indicated that different values of *a* have a certain impact on the calculation results of information entropy. However, based on the current analysis, it was unclear which value was the optimal result, and further analysis was needed. Despite this bias in the calculation results, the *JE* and *TC* curves exhibited similar trends. Specifically, the response of the three discretization methods was more consistent when the gauge combination corresponded to a larger *JE* value.

The difference in entropy-based indexes estimated by three discretization methods would lead to divergence in the optimization results. In this study, a total of 90,000 gauge combinations were randomly generated. Pareto solutions were then selected based on the optimization criterion of maximizing the *JE* (joint entropy) and minimizing the *TC* (total correlation) from these combinations. However, due to space constraints, only the Pareto solution results obtained from network optimization using the floor function-based discretization approach with *a* being 150 are presented in Figure 5. The Pareto solution results obtained using the other discretization methods are presented in Figure S3. Scatter points satisfying the max*JE*-min*TC* criterion were found to be distributed close to the Pareto curve, regardless of the discretization method used. However, there were differences in the number and information entropy index of the Pareto solution sets obtained using the different discretization methods. As illustrated in Figures 5 and S3, it can be observed that the majority of scattered points representing the Pareto solution set were located on the lower right boundary, which represented all potential combinations of candidate points. This represented a trade-off between the two objective functions: maximizing the joint entropy and minimizing the total correlation of the optimal rain gauge network. As indicated in Table 2, out of the potential 90,000 combinations, 27 gauge combinations derived using the floor function-based discretization method with *a* = 50 were identified as Pareto solution sets.

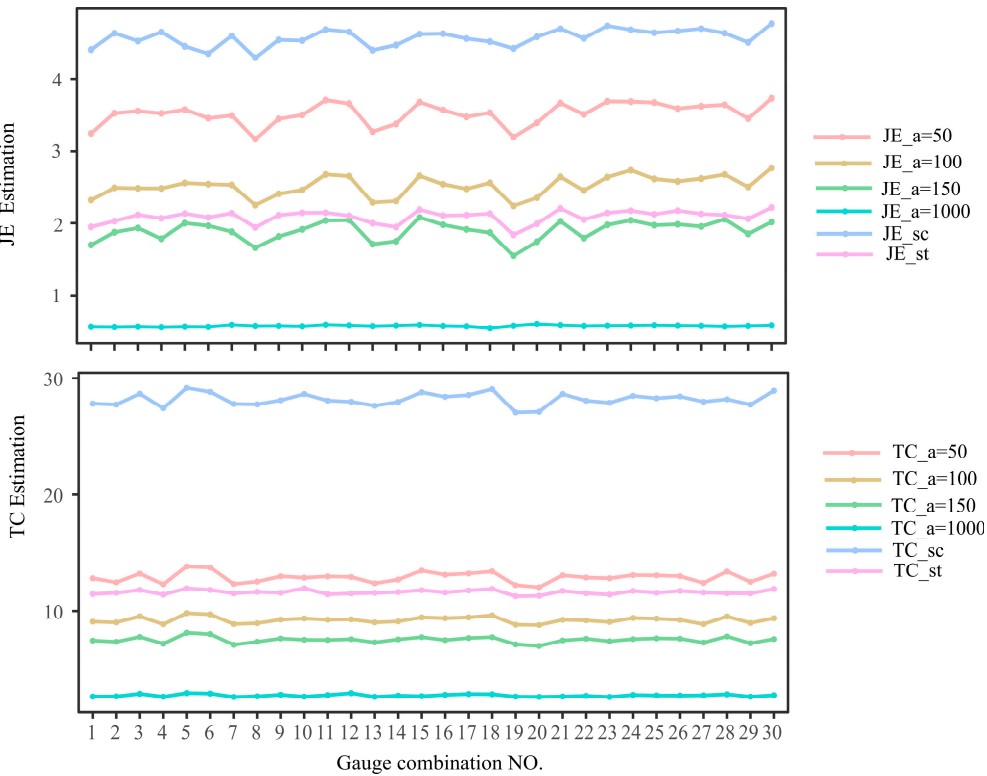

**Figure 4.** Comparison diagram of information entropy-based index (*JE*: Joint Entropy and *TC*: Total Correlation) estimated results.

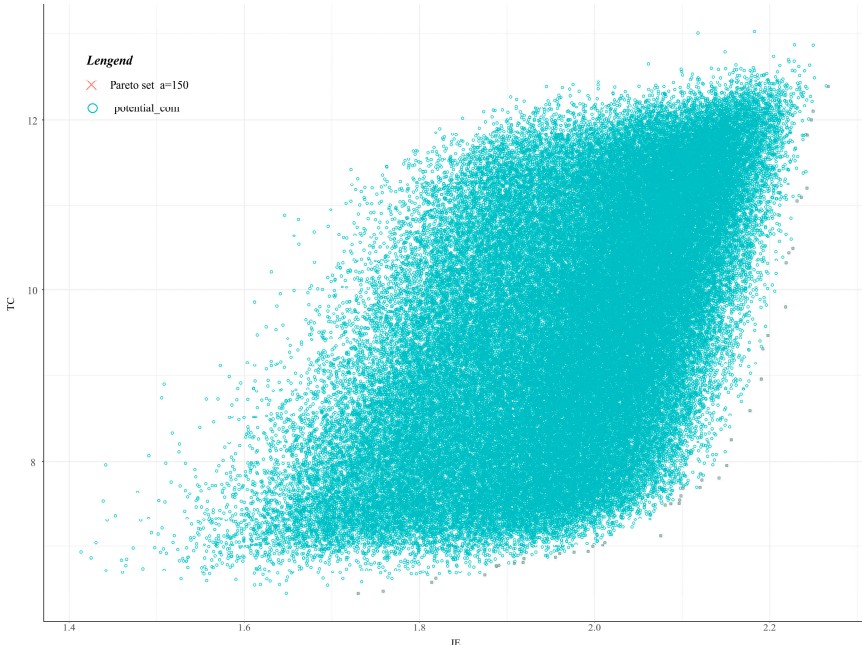

**Figure 5.** Process of optimization of network design based on the Pareto solution through flood function-based discretization approach with *a* being 150.

**Table 2.** The number of pareto sets for all discretization methods.

| Discretization Approaches | Number |
|---|---|
| Floor function-based with $a = 50$ | 27 |
| Floor function-based with $a = 100$ | 29 |
| Floor function-based with $a = 150$ | 48 |
| Floor function-based with $a = 1000$ | 100 |
| EWH-Sc | 17 |
| EWH-St | 37 |

A total of 29 and 48 gauge combinations yielded the Pareto solution sets for the discretization method using the floor function-based approach with parameter a values of 100 and 150, respectively, out of the potential 90,000 combinations. When the parameter a value of the floor function-based approach reached 1000, there were 100 different gauge combinations in the Pareto sets. This suggested that the number of potential Pareto solution sets was proportional to parameter a of the floor function-based approach. For the other two discretization methods, the number of Pareto sets was 17 for EWH-Sc and 37 for EWH-St.

Due to the large number of Pareto solution sets, we quantified the importance of each site's selection frequency in the Pareto solution set and displayed the optimization results in Figure 6. It can be seen from the figure that although different discretization methods influenced the final optimization results in general, the optimal networks obtained by all of the discretization approaches followed similar spatial distribution patterns: the frequency of incoming selection of stations at the basin boundary was high (such as stations S8, S43, S3, S17, and S6), which was crucial to the station network system; however, the frequency of site selection within the watershed was relatively low (such as sites S40, S12, S27, S29, and S26).

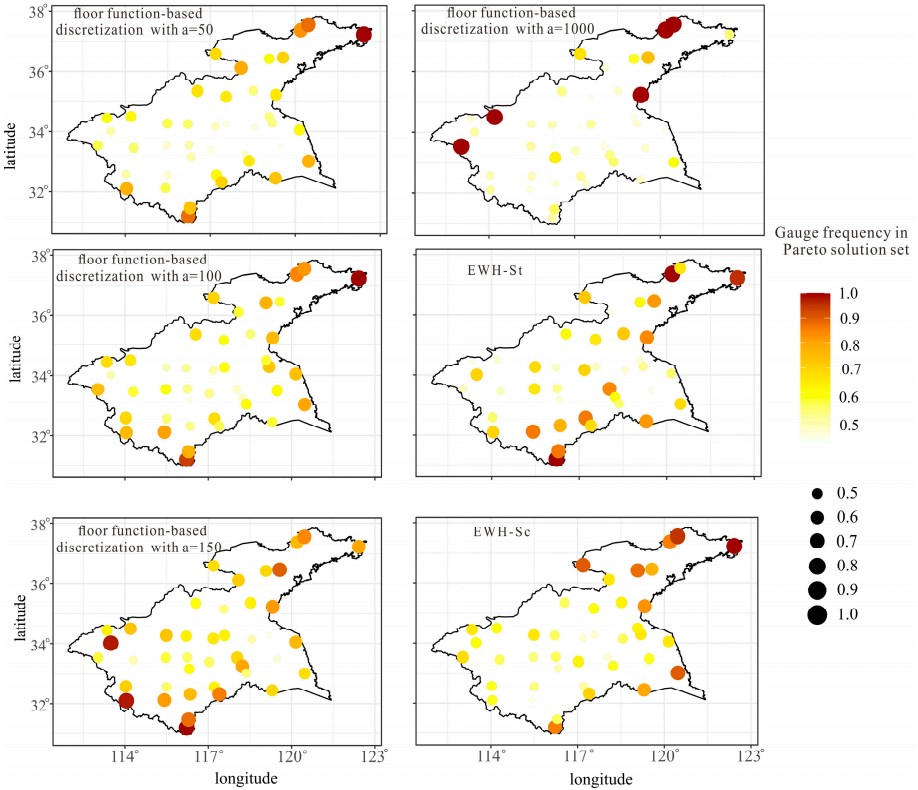

**Figure 6.** Comparison of station network optimization results using different discretization methods.

For the sake of different optimization results obtained using different discretization methods, this study calculated the sample standard deviation (SD) between the marginal entropy (*ME*) of 43 stations in the basin using different discretization methods. Here, we mainly analyzed the matching degree between the standard deviation series and marginal entropy series. It can be seen from Figure 7 that the matching degree between the marginal entropy series and the standard deviation calculated based on the Scott and Sturges discretization methods was significantly lower than that using the floor function-based method. For the floor function-based method, when parameter *a* = 100, the matching degree of the marginal entropy calculation results and the serial standard deviation was the best. This study intended to adopt the discretization method when parameter *a* of the floor function-based approach was 100 as the optimal discretization method for calculating the information entropy index.

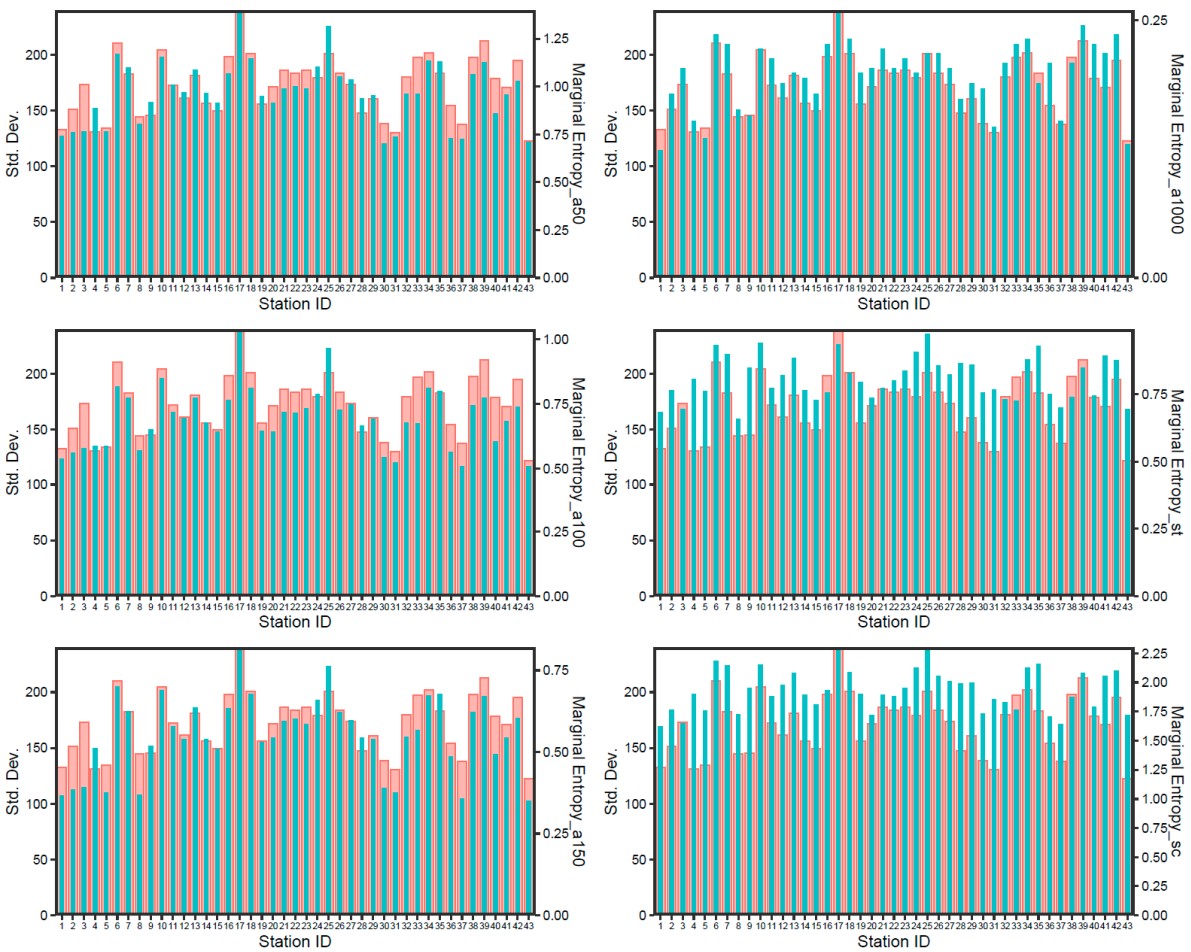

**Figure 7.** The establishment process of the optimal discretization method.

*4.3. The Impact of Time Variability in Precipitation on the Optimization Results of Gauge Networks in the Huaihe River Basin*

The average annual precipitation in the Huaihe River Basin (HRB) was 875 mm during 1968–2018, with 911 mm in the Huaihe River system and 788 mm in the Yishu River system. The precipitation was unevenly distributed in the region, being greater in the southern part than in the northern part, and in the mountainous area than in the plain, and in the coastal area than inland. The average annual precipitation in the southern Dabie Mountains ranged from 1400 to 1500 mm, while in the northern Yellow River coast, it ranged from only 600 to 700 mm. The interannual variation of precipitation was significant, with an average annual precipitation of 1185 mm in the entire basin in 1970 and only 578 mm in 1996. The distribution of precipitation within the year was uneven, with the rainy season

concentrated from May to September in the upper reaches of the Huaihe River and Huainan mountainous areas [24–27], while it was concentrated from June to September in other regions. The precipitation during the flood season (June–September) accounted for 50–75% of the annual precipitation. Regional flood, drought, urban waterlogging, and storm surge disasters have frequently taken place in the HRB in the last 60 years owing to its unique climate and surface conditions. The temporal variability in the regional precipitation in the HRB has been reported in the literature [28–32].

As a result of anthropogenic greenhouse gas emissions, the climate is changing and the composition of Earth's atmosphere is being altered, which has not only increased the complexity of hydrometeorological simulations but also necessitates the incorporation of trend-caused nonstationarity in modeling hydrometeorological variables [33–35]. In this study, trend-caused nonstationarity was first incorporated into the optimal design of the rain gauge network.

In order to determine whether the fluctuating nature of precipitation would impact the optimization outcomes of the network, Figure 8 displays the results of the Mann–Kendall trend tests conducted on all 43 stations. It is evident from Figure 8 that, in general, nearly 75% of the 43 chosen stations displayed nonstationary trends at a significance level of 5%.

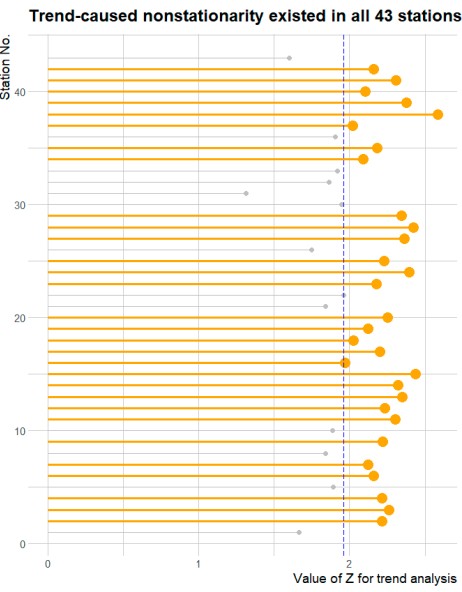

**Figure 8.** Lollipop chart of Mann–Kendall trend test results for the processed data in all 43 stations. Bold yellow lollipops represent the processed data of stations showed trend-caused nonstationarity at the significance level of 5%. The blue dashed line represents the threshold of Z (1.96) at the significance level of 5%.

In order to accurately quantify the impact of trend-caused nonstationarity on the results of station network optimization, this section employed the sliding window method to conduct the secondary analysis of the gauge network in the HRB. Specifically, a window width of 30 years (referred to as a sliding window, SW) was selected, and the precipitation subsequences in the HRB were chosen for re-analysis using multi-objective station network optimization. This optimization was carried out with the discretization means of the floor function-based approach, with a set to 100. From the results presented in Figure 9, it is evident that time-varying trends in the sequence will have a significant impact on optimization outcomes in the Huaihe River Basin. Specifically, stations S8 and S17 exhibited a higher frequency of selection, as indicated by the dark blue color in Figure 9 (approximately 0.95 for both stations). This suggested the greater importance of these two stations in the entire network. Generally speaking, a higher Z statistic was associated with a more pronounced time-varying characteristic for the selected frequency corresponding to each station in Figure 9 (i.e., greater color contrast).

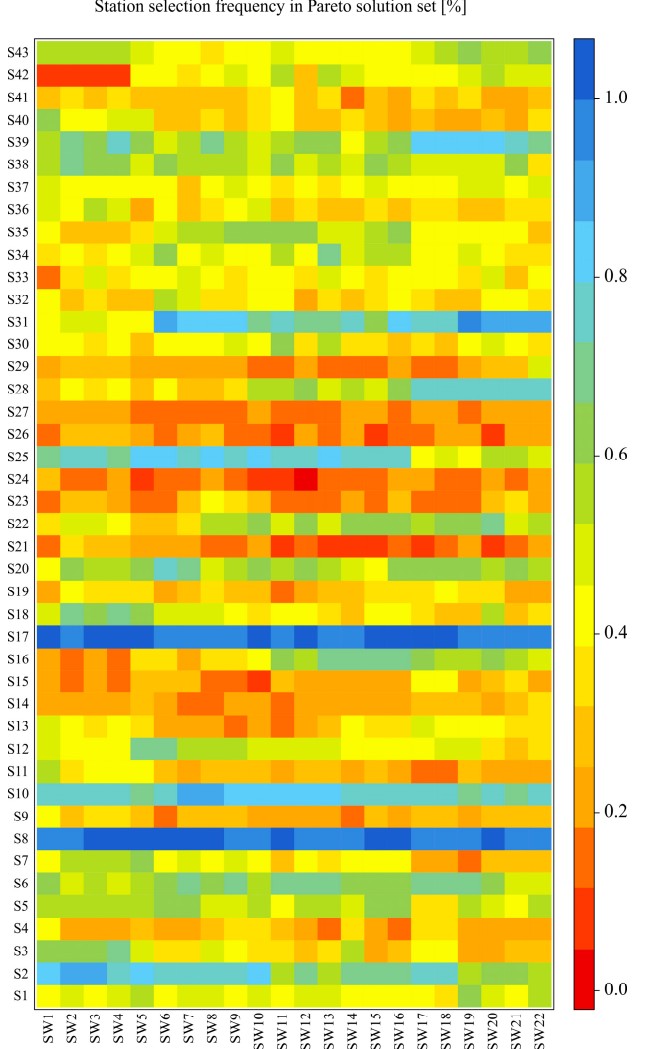

**Figure 9.** Time-varying analysis of selection frequency of stations in the Huaihe River Basin under the 30-year sliding window (SW) width condition.

Therefore, in future station network optimization analyses, it is necessary to carry out uncertainty research according to local conditions in view of climate change and human activities.

## 5. Discussion

The information entropy-based method for optimizing rain gauge networks is commonly used in the field of hydrometeorological station network optimization. However, uncertainties arise during the calculation process due to the high sensitivity of discretization methods. Additionally, the influence of spatiotemporal variability on the design optimization of hydrometeorological station networks has been widely recognized. When optimizing the design of such networks, it is crucial to consider the nonstationary nature of climate change, human activities, and hydrological processes. This is particularly important for rainfall events, which exhibit high heterogeneity, localization, and strong influences from geographical, topographical, and climatic factors. Different time periods may have varying impacts on the design outcomes for station networks, implying that the optimal layout may only be suitable for specific observation periods. Although previous researchers have conducted considerable work on station network optimization [3,12,17], dynamic analysis of hydrological and meteorological station network optimization in response to climate change is limited in terms of the nonstationarity of hydrometeorological time series.

In this study, we proposed an entropy-based approach for optimal precipitation monitoring network design in the Huaihe River Basin (HRB), China, considering trend-caused nonstationarity. The approach was applied to a rain gauge network containing 43 stations located in the HRB. Precipitation records of 51 years on a daily time scale were used for analysis. The main contribution of this study to the research of rain gauge network design lies in selecting the appropriate discretizing approach for calculating entropy-based indexes and quantifying the impact of trend-caused nonstationarity on the final results of the optimal network using the sliding window method. The final network scheme using a floor function-based approach with *a* = 100 obeyed the multi-objective optimization criterion, which proved the robustness of the proposed approach.

In order to verify the necessity of the data fusion method, comparisons of the matching degree between the standard deviation and marginal entropy from the processed and original precipitation series are shown in Figure 10. As shown in Figure 10, the matching degree between the standard deviation and marginal entropy from the processed data was relatively higher than that from the original precipitation series, which showed the advantage of data conversion in helping to quantify the information content more precisely.

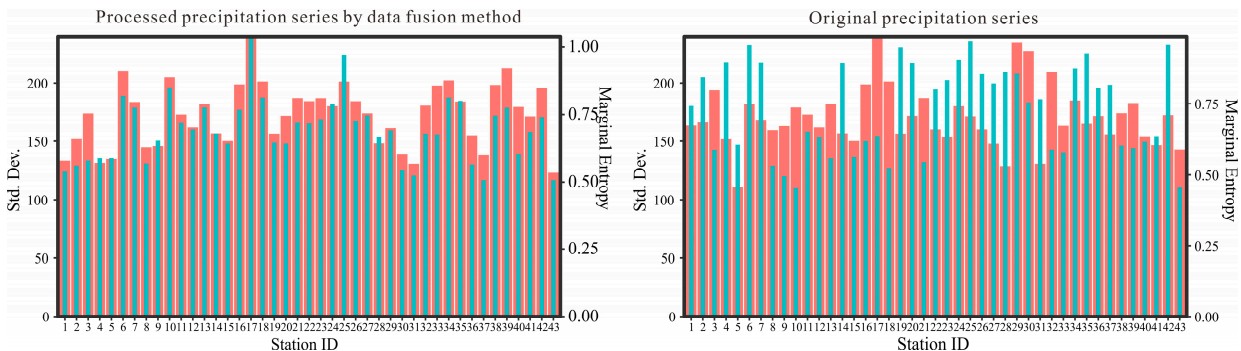

**Figure 10.** Comparison of the processed and original precipitation series for the matching degree between the standard deviation series and marginal entropy series.

## 6. Conclusions

An entropy-based framework incorporating trend-caused nonstationarity into hydrometeorological network design is developed in this study. According to its application in the Huaihe River Basin, the conclusions of this study can be drawn as follows:

(1) Careful selection of discretization technology is the basis for station network optimization. This study compared the network optimization results derived from three kinds of discretization methods, including the floor function-based approach, Scott's equal bin width histogram (EWH-Sc) approach, and Sturges's equal bin width histogram (EWH-St) approach (Figure 5). The floor function-based approach with *a* = 100 was selected as the most suitable discretization method for this study by optimizing the matching degree of the variance and edge entropy sequence of the measured values at each station.

(2) The criterion of maximizing the joint entropy and minimizing the total correlation (max*JE*-min*TC*) was able to generate potential Pareto solution sets for the optimal network. The frequency of selecting sites in the Pareto solution set proposed in this study provides a new approach for characterizing the results of station network optimization.

(3) Due to the trend-caused nonstationarity in almost 75% of all stations in the HRB, taking the impact of temporal variability in the precipitation series on the final rain gauge network optimization results into consideration is of great significance. The analysis results indicated that the degree of nonstationarity in the processed precipitation series is directly proportional to the frequency of station selection.

**Supplementary Materials:** The following supporting information can be downloaded at: https://www.mdpi.com/article/10.3390/w15173115/s1.

**Author Contributions:** J.L.: Conceptualization, Methodology, Writing and editing manuscript, Supervision. Y.L.: Formal analysis and Reviewing. Y.W.: Writing—original draft. P.X.: Model fitting and Supervision. All authors have read and agreed to the published version of the manuscript.

**Funding:** This study was supported by the National Natural Science Foundation of China (No. U22A20555, 41501053, and 42301026).

**Data Availability Statement:** Site-based meteorological data used in this article are derived from the National Meteorological Information Center, China Meteorological Administration. (http://data.cma.cn (accessed on 27 August 2023)).

**Acknowledgments:** Site-based meteorological data used in this article are derived from the National Meteorological Information Center, China Meteorological Administration. (http://data.cma.cn (accessed on 1 January 2020)).

**Conflicts of Interest:** The authors declare that they have no conflict of interest.

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
