# Peer review of "Utilizing Entropy-Based Method for Rainfall Network Design in Huaihe River Basin, China"

_water, doi:10.3390/w15173115_

Round 1

Reviewer 1 Report

A considerable volume of work seems to have been developed, but the manuscript is very badly written and the main messages are very confusing and poorly transmitted. The manuscript requires to be rewritten with a deep work of major revisions and editing service. Reviewer overall recommendation is that the manuscript should be deeply revised and resubmitted, after assuring minimal acceptable conditions to be read, revised and evaluated by peers. The Abstract, the Introduction, and Section of Results Section, in particular, need to be largely improved.

Some few specific examples of possible improvements/corrections:

Lines 10 to 12 - It is written: “The nonstationary characteristics caused by the significant trend of hydrometeorological series in the context of climate change will inevitably have a certain impact on the optimization results of the gauging network.” It is intended to be “caused by the significant variation”, instead of “caused by the significant trend” and “have a certain impact on the selection of the” instead of “have a certain impact on the optimization results of the”?

Line 12 – Perhaps “This study proposes”, instead of “This study proposed”? Verb tense seems more suitable in the present, at least here, in the Abstract. However, comparable remarks/corrections can be pointed in several instances along the text of the manuscript.

Lines 14 to 17 – Perhaps “the main goals of this study” could be better expressed, even if briefly, in these Lines, and the grammar corrected/improved. It is not clear what are exactly the main goals and what are exactly the tools used to achieve them. A clear distinction should be made between the goals and the tools used.

Lines 17 to 23 – The first sentence is too much long and both sentences presented in these Lines are very confusing and badly written.

Lines 29-30 - It is written: “The key elements involved in hydrological cycle and water resource management activities” Key elements for what?

Lines 31-32 - The smooth and efficient operation of what? Of the mentioned key elements “such as rainfall, runoff, sediment, water quality, and water environment”? Do this make sense? How is (or can be) the rainfall operated, for example?

Line 79 – Perhaps “are potential factors” instead of “is potential factors”?

Lines 104-105 – Perhaps “The distribution along the year of the annual precipitation is uneven”, or “The distribution of monthly precipitation is uneven, or similar, instead of “The distribution of annual precipitation is uneven”.

Figure 1 - Meaning of the acronym DEM shown in Figure 1 is missing and should be mentioned in the text. Does it mean “digital elevation model”? Do the colors in Figure 1 represent the surface elevations in meters, with an important part of the basin apparently bellow the sea level (up to -12 m)?

Lines 151 and 153 - The base of the logarithms present in Equations 1 and 2 should be explicitly given/stated. For example, is the base 10, 2, or any other?

An optimization problem is systematically mentioned along the manuscript, but such problem is never exactly described nor fully and clearly formulated.  What are exactly the hydrological variables that are intended to be measured in the stations and analyzed in this work and that are often just vaguely mentioned along the manuscript. It is intended to be simply the daily precipitations or groupings of daily precipitations? Just in the Lines 159 to 167, Equation 4 included, an attempt for describing the optimization problem finally seems to be indirectly schematized/sketched. What are exactly the decision variables? Just the number m and corresponding location of the rain gauge stations to be chosen/selected between a full set of stations in number of d already existing stations in the river basin. The optimization problem should be clearly and fully defined, and also briefly explained since the first time it is mentioned in the text, or, at least, in the Introduction of the manuscript.

Lines 169-170 - It is written: “The above entropy-based metrics are calculated in the discrete form rather than using a continuous form. This study discussed three kinds of discretization approaches: …” The first sentence seems absolutely redundant and thus unnecessary. The study discussed when and where? Perhaps “This study discusses” instead of “This study discussed”? Also, the discretization approaches are required for what? The number and location of the existing stations and the daily precipitations are not already discrete variables completely defined? Additionally, it is not clear what is exactly the relation between the optimization problem under consideration and the 3 discretization approaches presented in this subsection 3.2.

Line 189 and followings – 4. Results - The current section of results seems generally presented in a form disconnected, and too much casuistic, and these aspects need to be naturally corrected and improved.

Line 199-201 – Why in Figure S1 just 16 stations are selected from a total of 43, and in the Table 1 the total number of selected stations is 24?

Line 215 – Caption of Table 1 - Perhaps “of the grouped precipitation data”, or similar, instead of “of the data”?

Lines 220 to 222 – Why just 200 gauge combinations and how the stations were exactly selected? It is not clear any possible relation of this number with the mentioned “three kinds of quantization methods” considered. The total number of feasible combinations of the discrete combinatorial optimization problem should be clearly and explicitly presented, and this number of 200 should be discussed in such context. Also, why just 30 combinations are presented in Figure 3, and 90000 gauge combinations are mentioned in the Line 239, and how were they exactly selected/chosen.

Line 347 - Discussion and Conclusions should be presented in two separate sections.

The language and the presentation of the scientific contents need to be largely improved.

Author Response

Great appreciations to your constructive suggestions and advice to our manuscript. We have made a systematic revision to the revised version and responded to the comments suggested by three reviewers one by one. And the detailed information is shown as follows.

Reviewer 2 Report

This paper examined entropy-based data processing for rain gauges in China, which is an interesting approach but needs to clarify the detailed goal and the application of the proposed method.

I am not quite sure about the advantage of data conversion in this way. So, what is the analysis result of the precipitation of the study area? The authors should present how this method helps the analysis of the characteristics, patterns, or trends of the long-time precipitation data in the study area.

Author Response

The detailed response to the comments of reviewer 2 has been presented in the attachment.

Reviewer 3 Report

Dear Authors,

This is a good and carefully written manuscript, combining important practical problems of rainfall network design and associated metabolism with entropy-based indexes. The manuscript is clear and consist but scientifically relevant.  

The manuscript can be published after necessary editorial improvements.

Sincerely yours,

Reviewer

Minor editing of English language required

Author Response

Response to reviewer 3

Minor editing of English language required

Response: we have tried our best to polish the words and sentences all over the manuscript.

Round 2

Reviewer 1 Report

Please refer to the first detailed and constructive Review Report of Reviewer # 1.  Some Reviewer # 1 Recommendations were fully addressed and the corresponding aspects of the manuscript presentation have in fact improved, but others not. Authors should understand that a scientific text is (should be), as much as possible, clear, accurate, and rigorous/demanding. A lack of scientific clearness and accuracy seems latent along the entire manuscript, with a wording and claims that seem to not have not correspondence nor in the real research developed nor in the results achieved. Some aspects of the manuscript presentation, particularly related with the optimization procedures and optimization claims, for example, continue to be not clear nor scientifically acceptable. Reviewer cannot recommend the publication of this manuscript in this form nor in any similar form, unless these issues are completely solved. Current Sections 3, 4 and 5, in particular, need to be largely improved.

The language style and clearness and several editing aspects need to be improved.

Author Response

 In the revised manuscript, contents in Sections 3, 4 and 5 have been improved and polished through our efforts.

In order to present the optimization procedures clearer, we plotted the flowchart of this study in Figure 3.
